# Qualitative analysis of stakeholder interviews to identify the barriers and facilitators to the adoption of point-of-care diagnostic tests in the UK

Jeremy R Huddy ,[1] Melody Zhifang Ni ,[1] James Barlow,[2] George B Hanna[1]

[1]Department of Surgery and Cancer, Imperial College London, London, UK
[2]Imperial College Business School, Imperial College London, London, UK

**Correspondence to**
Professor George B Hanna; g.hanna@imperial.ac.uk

## ABSTRACT

**Objectives** This study investigated the barriers and facilitators to the adoption of point-of-care tests (POCTs).

**Design** Qualitative study incorporating a constant comparative analysis of stakeholder responses to a series of interviews undertaken to design the Point-of-Care Key Evidence Tool.

**Setting** The study was conducted in relation to POCTs used in all aspects of healthcare.

**Participants** Forty-three stakeholders were interviewed including clinicians (incorporating laboratory staff and members of trust POCT committees), commissioners, industry, regulators and patients.

**Results** Thematic analysis highlighted 32 barriers in six themes and 28 facilitators in eight themes to the adoption of POCTs. Six themes were common to both barriers and facilitators (clinical, cultural, evidence, design and quality assurance, financial and organisational) and two themes contained facilitators alone (patient factors and other (non-financial) resource use).

**Conclusions** Findings from this study demonstrate the complex motivations of stakeholders in the adoption of POCT. Most themes were common to both barriers and facilitators suggesting that good device design, stakeholder engagement and appropriate evidence provision can increase the likelihood of a POCT device adoption. However, it is important to realise that while the majority of identified barriers may be perceived or mitigated some may be absolute and if identified early in device development further investment should be carefully considered.

## INTRODUCTION

Point-of-care tests (POCT) are tests that are carried out in close proximity to patients by non-laboratory trained personnel either in healthcare settings or the home. Technological advances have led to more tests traditionally performed in clinical laboratories being available to clinicians at the bedside.[1] This convenience can allow for real time decision making within patient care pathways and has led to improved access to diagnostics, reduced time to treatments, patient satisfaction and cost savings.[2] Further advantages include

---

### Strengths and limitations of this study

► Semistructured interview study incorporated a diverse range of views from stakeholders involved in the implementation of point-of-care test (POCT).

► Thematic analysis identified the barriers and facilitators that should be considered when developing POCT.

► Interview topic guide grouped all POCT together and incentives and disincentives may vary between different POCT categories (eg, patient self-testing, testing in primary care, etc).

---

more accurate clinical decision making regarding when to initiate treatments and improved guidance for patients regarding when to seek advice or visit a healthcare professional.

Healthcare systems represent significant challenges to technology adoption. It is often not failures in the technology but cultural, organisational and financial aspects of the system that prevent implementation. The diffusion of innovation traditionally follows an S-shaped curve as an innovation passes through stages of adoption and spread before plateauing at mainstream adoption.[3] [4] However, the length of time taken innovations to be adopted in health systems has been well documented around the world; one review suggests an average of 17 years.[4] Efforts need to be made to speed up this process and identify factors that can hamper or accelerate the adoption journey so that the benefits to patients and society of these devices can be realised.

Barriers and facilitators can be reliably studied using qualitative techniques.[5–7] A better understanding of the adoption motivations for POCTs may inform strategies to mitigate barriers and promote facilitators that expedite the time it takes for devices to become integrated into patient pathways.

Barriers and facilitators to healthcare technologies were investigated as part of the 2016 Accelerated Access Review (AAR) commissioned by the UK government and supported by the Wellcome Trust).[8] This review began with an interim report that highlighted a list of barriers that stand in the way of innovation adoption in healthcare systems. However, the findings from this report reflect a wide variety of healthcare technologies including diagnostics, medical devices and pharmaceuticals findings and therefore may not represent the specific challenges of expediating the development and adoption of POCT technology. Given the increasing number of POCTs available and their potential impact across clinical pathways there is value in investigating the specific barriers and facilitators to POCT technology adoption. This would allow better understanding of the potential challenges in commercialising and implementing POCT tests, so that modifications may be made early in their development and guide the collection of evidence needed to address the needs of clinical and other stakeholders. Two systematic reviews have previously investigated such barriers; in their systematic narrative review of the barriers to hospital-based adoption of POCT Quinn *et al* identified the most prevalent barrier categories being the economics of adoption, quality assurance and regulatory issues, device performance and data management and staff and operation issues.[9] Jones *et al* investigated the attitudes of primary care clinicians in Europe and Australia towards POCTs and reported the three main barriers being related to the impact on clinical practice, impact on patient-clinician relationships and perceived patient experience.[10]

Previous work has led to the development of the Point-of-Care Key Evidence Tool (POCKET) tool[11] that reports the evidence that is a required by different stakeholder groups to facilitate good decision making in regards the adoption of POCT. This has demonstrated that the evidence is multidimensional and includes evaluation of both the test and the test device itself. Currently, there exists a shortfall in available translational evidence compared with what is required by decision makers.[12] As part of this study a series of in-depth semistructured interviews was undertaken with a broad range of relevant stakeholders. The aim of this study was to undertake a qualitative analysis of these interviews with a specific focus on identifying the barriers and facilitators that exist to the adoption of POCTs within patient pathways.

## METHODS

A qualitative analysis of the 41 semistructured stakeholder interviews that were conducted to develop POCKET was undertaken using grounded theory. Two interviews were undertaken with two participants concurrently at the participant's request. The study description and results are summarised in accordance with the Consolidated Criteria for Reporting Qualitative Research checklist.[12] Stakeholders included were clinicians (including laboratory staff and members of trust POCT committees), commissioners, industry, regulators and patients. The methodology for the development of the POCKET checklist including the interview phase, topic guide and recruitment strategy has been previously published.[13 14] Written consent was obtained from all participants. The interviews were undertaken by JH. The interview participants received information relating to the study in advance of the interview through a participant information leaflet. The interviews were semistructured and a topic list was prepared in advance and piloted (online supplemental appendix 1). Interviews were tailored to the role and level of experience of the interviewee. This ensured the interviews remained relevant to the different stakeholders. The interview topic guide included a section relating to adoption where barriers and facilitators were explored. Interviews were undertaken either in person, by telephone or through Skype (Microsoft, Seattle, Washington, USA). Interviews were recorded. Following verbatim transcription transcripts were not returned to participants for verification. Interviews were analysed by two researchers (JH and a clinical research nurse working within the National Institute for Health Research (NIHR) Diagnostic Evidence Co-operative London) who independently reviewed the transcripts using a constant comparative technique before meeting to compare emergent themes. Interviews were undertaken until saturation had been reached, as demonstrated by the absence of new themes emerging from the analysis. NVivo (V.11.2.2 QSR International) was used to query the data and provide matrix coding so that the frequency of stakeholders in each group who identified a particular barrier or facilitator could be determined.

## RESULTS

Forty-three stakeholders comprising regulators (10), Industry (9), commissioners (8), clinicians (8) and patients (8) were interviewed between August 2014 and December 2015. Recruitment rates from invitation were 80% for clinicians and 29% for commissioners. Regulators, industry representatives and patients were recruited through bodies external to the study group and recruitment rate is not known. Twenty-three (55%) interviews were undertaken face to face and 19 (45%) by Skype or telephone. Interviews had a median length of 31 min (range 12–58 min). For the purpose of reporting interviewees are referred to as stakeholders.

Thematic analysis highlighted 32 barriers in six themes and 28 facilitators in eight themes to the adoption of POCTs. Six themes were common to both barriers and facilitators (clinical, cultural, evidence, design and quality assurance, financial and organisational) and two themes contained facilitators alone (patient factors and other (non-financial) resource use). Tables 1 and 2 summarise the barriers and facilitators identified within each theme and the number of participants from each stakeholder group who identified each. The sections below

**Table 1** Identified barriers to the adoption of POCT and frequencies of stakeholders who highlighted each by stakeholder group

| Theme | Identified barrier | Comm. | Clin. | Reg. | Industry | Patients | Total |
|---|---|---|---|---|---|---|---|
| Clinical | Inappropriate use of test | 1 | 1 | 1 | 0 | 1 | 4 |
| | Lack of clinical utility | 1 | 4 | 0 | 2 | 0 | 7 |
| | Reduced accuracy | 0 | 2 | 2 | 0 | 0 | 4 |
| | Discourages use of clinical judgement | 0 | 1 | 0 | 1 | 0 | 2 |
| Cultural | Lack of awareness | 0 | 0 | 0 | 2 | 0 | 2 |
| | Lack of clinician buy-in | 0 | 0 | 0 | 6 | 1 | 7 |
| | Resistance to change | 4 | 0 | 1 | 2 | 0 | 7 |
| | Need for service reconfiguration | 5 | 0 | 0 | 2 | 0 | 7 |
| | Lack of Accountability | 0 | 0 | 0 | 0 | 1 | 1 |
| Evidence | Cost of evidence generation | 0 | 1 | 0 | 2 | 0 | 3 |
| | Lack of evidence | 2 | 0 | 1 | 0 | 0 | 3 |
| | Lack of evaluation methodology | 0 | 1 | 0 | 1 | 0 | 2 |
| | Repetition of small scale local studies | 0 | 0 | 0 | 3 | 0 | 3 |
| | Rate of technology development | 0 | 0 | 1 | 0 | 0 | 1 |
| Financial | Commissioning | 1 | 0 | 0 | 0 | 3 | 4 |
| | Cost of test | 5 | 3 | 1 | 3 | 2 | 14 |
| | Lack of investment in new technology | 0 | 1 | 0 | 3 | 0 | 4 |
| | Payer identification | 1 | 0 | 0 | 2 | 0 | 3 |
| | Reimbursement | 0 | 0 | 2 | 0 | 0 | 2 |
| | Siloed budgets | 1 | 1 | 1 | 1 | 0 | 4 |
| Organisational | Asset control | 0 | 0 | 1 | 0 | 0 | 1 |
| | Duration of consultation | 1 | 2 | 1 | 0 | 0 | 4 |
| | Increased workload | 2 | 1 | 0 | 1 | 0 | 3 |
| | Threat to laboratory services | 4 | 0 | 1 | 3 | 0 | 8 |
| | Reduced throughput | 0 | 0 | 1 | 0 | 0 | 1 |
| | Regulation | 0 | 0 | 0 | 2 | 0 | 2 |
| | Trust Autonomy | 0 | 0 | 0 | 2 | 0 | 2 |
| Design and Quality Assurance | Maintenance requirements | 1 | 0 | 1 | 0 | 0 | 2 |
| | Training burden | 2 | 6 | 4 | 2 | 0 | 14 |
| | Poor usability | 0 | 0 | 1 | 1 | 0 | 2 |
| | Lack of connectivity | 1 | 0 | 0 | 1 | 0 | 2 |
| | Lack of technical support | 0 | 0 | 1 | 0 | 0 | 1 |

Clin, clinicians; Comm, commissioners; Reg, regulators.

qualitatively summarise the themes that emerged during the interviews along with verbatim quotations.

### Clinical barriers and facilitators

The benefits that POCTs are able to bring to clinical pathways were widely recognised by all stakeholders. Particular focus was given to the communication benefits, better patient care, improved turnaround time and effective patient triage. However, there was an appreciation that technological advances may have led to some POCTs being available without or before a clinical need or pathway benefit could be clearly identified. Furthermore, even when a clinical need was determined, given the volume and accuracy offered if a laboratory was able to deliver a test result at a time point that doesn't affect the efficacy of the associated intervention, then the POCT's utility can still be questioned (unless other benefits can be demonstrated). Some stakeholders were concerned by the inappropriate use of POCTs, for example, patients self-testing too often or when not indicated by clinicians. POCTs were seen by some to discourage the use of clinical judgement, leading to an over-reliance on guidelines and treatment algorithms; conversely others described the improvements POCTs offer to doctor–patient communication, for example, when providing an objective test

**Table 2** Identified facilitators to the adoption of POCT and frequencies of stakeholders who highlighted each by stakeholder group

| Theme | Identified facilitator | Comm. | Clin. | Reg. | Industry | Patients | Total |
|---|---|---|---|---|---|---|---|
| Clinical | Improving doctor–patient communication | 2 | 0 | 0 | 0 | 0 | **2** |
| | Improving patient care | 0 | 2 | 1 | 2 | 0 | **5** |
| | Time to result | 0 | 4 | 1 | 2 | 0 | **7** |
| | More effective triage | 0 | 0 | 0 | 1 | 0 | **1** |
| | Reduced time to treatment | 4 | 0 | 5 | 1 | 0 | **10** |
| Cultural | Word of mouth | 0 | 0 | 0 | 0 | 4 | **4** |
| | Incentives to adopt | 0 | 1 | 0 | 1 | 0 | **2** |
| | Key opinion leaders | 1 | 0 | 0 | 0 | 2 | **3** |
| | NHS and industry partnerships | 1 | 1 | 0 | 1 | 0 | **3** |
| Evidence | Availability of patient-level data | 1 | 0 | 1 | 0 | 0 | **2** |
| | Successful pilot data (local level) | 0 | 0 | 0 | 2 | 0 | **2** |
| Financial | Cost savings | 1 | 0 | 1 | 1 | 0 | **3** |
| | Improved efficiency | 3 | 2 | 1 | 2 | 0 | **8** |
| Organisational | Connectivity | 0 | 0 | 0 | 0 | 2 | **2** |
| | Flow | 0 | 0 | 0 | 0 | 2 | **2** |
| | Government policy | 0 | 0 | 0 | 1 | 0 | **1** |
| | POCT use in local enhanced services | 3 | 0 | 0 | 0 | 0 | **3** |
| | Locality of diagnostics | 0 | 2 | 0 | 2 | 0 | **4** |
| | Support from central pathology laboratories | 2 | 1 | 1 | 1 | 0 | **5** |
| Design and Quality Assurance | Good design and functionality | 0 | 1 | 0 | 1 | 0 | **2** |
| | Usability | 1 | 0 | 1 | 0 | 0 | **2** |
| Patient Factors | Patient demand | 0 | 1 | 1 | 0 | 0 | **2** |
| | Patient experience/patient journey | 3 | 1 | 2 | 2 | 0 | **8** |
| | Patient ownership | 3 | 0 | 1 | 1 | 7 | **12** |
| Resource Use | Reduced attendance to healthcare services | 6 | 1 | 1 | 0 | 0 | **8** |
| | Reduced hospital admissions | 0 | 0 | 1 | 1 | 0 | **2** |
| | Reduced length of stay | 0 | 0 | 1 | 1 | 0 | **2** |

Clin, clinicians; Comm, commissioners; NHS, National Health Service; POCT, point-of-care test; Reg, regulators.

to justify why antibiotics are not required for self-limiting viral illness. Verbatim quotes to highlight the clinical barriers and facilitators to POCT adoption are provided in box 1.

### Cultural barriers and facilitators
Many stakeholders felt that there still exists a lack of awareness to the POCT technology that is currently available and their potential benefits. Furthermore, a particular frustration from industry and patients was the lack of engagement with POCTs by the clinical profession. POCT technology was recognised to require a fundamental cultural change in clinical practice. Driving such change was felt to require widespread incentivisation that was usually financial with examples cited in the section relating to Financial Barriers and Facilitators.

Despite such challenges, it was noted that enthusiasm from those with experience of POCTs was contagious, and local momentum was seen as a significant enabler.

*"it is a challenge to deliver equivalent analytical accuracies you can achieve within the main laboratory"* (R)
*"clinicians may use the tests for other conditions for which they're not evaluated"* (CI)
*"there may also be kind of a tendency where doctors may rely more on the diagnostic test than on their own clinical skills"* (CI)*"*
*"I think that point of care offers the opportunity for clinicians to, if correctly used, to improve the quality of care of their patients"* (I)

Key opinion leaders and early adoption sites were seen to actively contribute towards adoption as were industry and National Health Service (NHS) partnerships. Verbatim quotations to highlight the cultural barriers and facilitators to POCT adoption are provided in box 2.

### Evidence barriers and facilitators

The evidence required to facilitate POCT adoption has been described in the POCKET checklist[12] but two particular components were highlighted by stakeholders that enable adoption at a local level. First, the availability of patient-level data to demonstrate that the implementation of a POCT can actually make a difference to patient care within a population and second, local pilot studies to show that a POCT programme can actually work in a particular local practice. However, stakeholders commented that such examples were often not shared, and improvements were needed in the ability of healthcare organisations to learn from the successes, and failures, of others. This was especially felt by industry stakeholders who criticised the need for repetitive local evaluations assessing impact when this had already been demonstrated in equivalent populations or health service contexts.

The challenges of effective evidence generation in POCT technology are universal to all in vitro diagnostics and were frequently commented on; namely the cost of evaluation, the length of time high quality studies take to undertake at a time when there is a rapid rate of technological development and the methodological gaps.

*"more supportive and less distrustful of point of care devices"* (I)
*"Doctors are the most reluctant group, in many ways, to change something, and to change a habit"* (Co)
*"the biggest single barrier is endorsement and support by medical clinicians"* (P)
*"NHS and industry have to work together"* (I)
*"I see change but I see it over a period of years and not weeks and months"* (Co)

*"you need a second raft of evidence that the intervention actually makes a difference to patient care or patient perception of care"* (Co)
*"we are very keen to try and get good examples of how it* [POCT strategies] *can actually work in real practice in a local environment"* (I)
*"there is no properly well established methodology like you would have for drugs"* (I)
*"that everyone in the UK wants to repeat what's already been done"* (I)
*"money, time to evaluate them properly, understanding where they fit in the pathway, whether it fits in with the tariff system, and I think a genuine lack of understanding of how you demonstrate health gain in a sustainable way"* (Co)

Verbatim quotations to highlight the evidence barriers and facilitators to POCT adoption are provided in box 3.

### Financial barriers and facilitators

The portrayal of POCTs as being costly was a significant barrier to their adoption. While, as one commissioner described, this may be 'perceived', the increased cost of POCT devices on a test-by-test basis caused a front loading of cost that was unattractive to payers. The presence of budget silos within the NHS was frequently cited as a barrier particularly relevant to POCTs as there is often a mismatch between who funds the test and who saves as a consequence of implementation. Furthermore, identifying the payer within an organisation is frequently unclear, causing uncertainty within industry regarding whom to approach. This was compounded by a lack of appropriate reimbursement schemes for diagnostics, particularly in primary care, leading to a disincentive to adopt POCTs as laboratory diagnostics have already been purchased through contracted agreements with a local trust.

Challenges were highlighted in engagement with clinical commissioning groups (CCG) regarding diagnostics with the current national set up allowing for geographical variation. A particular example of this was POCT anticoagulation testing for patients taking warfarin. Conversely, at a local CCG level POCTs were identified in contributing to establishing an increasing number of local enhanced services, for example POCT D-dimer measurement for patients with suspected deep vein thrombosis. Overall, lack of investment in new technology was a clear frustration to industry and there was awareness of the need for decision makers to recognise the extensive societal cost savings that POCT investment can bring, such as reducing the spread of infectious diseases.[14] Verbatim quotations to highlight the financial barriers and facilitators to POCT adoption are provided in box 4.

### Organisational barriers and facilitators

The benefits of POCT technology to healthcare systems such as the NHS were clear in the interviews. They

> **Box 4  Verbatim quotations to highlight the financial barriers and facilitators to POCT adoption (P = Patient, I = Industry, Co = Commissioner, R = Regulator)**
>
> "*point of care testing is perceived as more expensive*" (Co)
> "*an additional cost or inconvenience or a burden (…) and an extra thing they* [healthcare professionals] *have to manage*" (P)
> "*it's often the barriers over the initial cost*" (R)
> "*who pays for it, and who sees the reward*" (Co)
> "*you have to go very high in the trust where it's almost like a triangle of where the budget lies*" (I)
> "*why invest more money into putting loads of evidence behind a new technology when history shows it might not get adopted*" (I)
> "*There are huge savings for the healthcare system. I think the big advantages are about patient flow and patient journeys*" (I)

included (1) integrating POCT systems into information technology systems so that results can be shared between patients at home, primary and secondary care; (2) improving patient flow through the availability of rapid results for decision making and (3) providing rural practices with access to local diagnostic services.

The relationship between clinical laboratory services and POCT sites was a contentious issue. Many stakeholders cited the laboratories as a barrier to POCT adoption given the perceived threats to their workload and income. However, there was an equal cohort of participants that saw the role of clinical laboratory services as facilitating the use of POCTs by contributing their expertise in asset management, quality assurance and training. There were particular examples of good practice from Cambridgeshire and Oxfordshire of laboratories supporting the appropriate use of POCTs in the community in the same way point-of-care committees act in hospital trusts.

Other organisational barriers highlighted were the difficulty in keeping control of assets within organisations as POCTs became more available and widespread, the challenges of reconfiguring services to accommodate the increased length of consultations that incorporate POCTs and the increased workload they can bring. Hospital trust autonomy represented a challenge for industry, who described the absence of a top down adoption policy for devices in the NHS, increasing the investment required to see their devices widely adopted. National government policy was cited as a means to improve the use of POCTs in the NHS. Industry participants also felt threatened by the changing landscape for device regulation including new European Union Regulations for medical devices and in vitro diagnostic medical devices.[15] Since the interview study was undertaken these concerns are likely to have been exacerbated by the added uncertainty caused by the ongoing negotiations for the UK leaving the European Union. Verbatim quotations to highlight the organisational barriers and facilitators to POCT adoption are provided in box 5.

> **Box 5  Verbatim quotations to highlight the organisational barriers and facilitators to POCT adoption (I = Industry, Cl = Clinician, Co = Commissioner)**
>
> "*the lab services look at point of care testing as being a bit of a threat.*" (Co)
> "*the laboratories are very reluctant to let go*" (I)
> "*they are starting to work up their knowledge of how they might be able to support appropriate point of care testing in different settings, so that their mindset is changing*" (Co)
> "*If I see a patient every ten minutes, how on earth am I going to get a point of care test in unless it delivers its results within a couple of minutes*" (Cl)
> "*balance between how long it [POCT] takes and how many patients you could see*" (Cl)
> "*I think back in the old days when the NHS was one organisation and you could just say 'right everyone does this' then you could get away with it but I don't think we can do it anymore*" (I)

### Usability and quality assurance barriers and facilitators

Quality assurance was recognised as vital for patient safety and clinical governance but presented significant challenges to successful adoption. Highlighted barriers were the maintenance required, training for multiple users, incompatibility with local information technology systems and an absence of technical support services. Usability was closely related to quality assurance as good design was felt to reduce training requirements and make devices safer to use. Improvements in this respect, especially when providing better functionality, facilitated adoption and diffusion. Verbatim quotations to highlight the design and quality assurance barriers and facilitators to POCT adoption are provided in box 6.

### Patient factors facilitators

Many stakeholders emphasised the popularity of POCTs to patients. POCT strategies were often seen to improve the patient journey and provide patients with a better healthcare experience. This has led to patient demand for the implementation of POCTs in many clinical pathways, often encouraged by media coverage. In the management of chronic disease the ability of POCTs to give patients control of their own healthcare was a significant benefit of the technology. All these factors were seen as enablers to the use of POCTs and no patient associated barriers were identified. Verbatim quotations to highlight

> **Box 6  Verbatim quotations to highlight the usability and quality assurance barriers and facilitators to POCT adoption (R = Regulator)**
>
> "*doctors believe that they're endowed by God to be able to perform analytical measurements without any training or need for any quality assurance unfortunately this is simply not the case*" (R)

   Huddy JR, *et al. BMJ Open* 2021;**11**:e042944. doi:10.1136/bmjopen-2020-042944

**Box 7** Verbatim quotations to highlight the patient factor facilitators to POCT adoption (P = Patient, CI = Clinician, R = Regulator)

"*patients themselves can see the advantage of it* [POCT]*"* (R)
"*it's quite difficult to say no to patients when they want a test*" (CI)
"*for me, it's been the most invigorating experience"* (P)

the patient related facilitators to POCT adoption are summarised in box 7.

### Other (non-financial) resource use facilitators

This final theme covers the reduction in patient–healthcare interactions that POCTs can lead to, namely a reduction in outpatient appointments, hospital admissions and length of stay. No barriers were identified relating to other (non-financial) resource use. Verbatim quotations to highlight the other (non-financial) resource use facilitators to POCT adoption are provided in box 8.

### DISCUSSION

This study has identified 32 barriers and 27 facilitators to the adoption of POCT technology. It is unlikely that all of these will apply to one particular device but will be dependent on factors such as the population, clinical pathway and experience of user. Many of the barriers and facilitators identified are closely related, for example, the cost of the test against potential pathway cost savings. This implies that some of the barriers may be perceived rather than absolute and therefore often can be readily overcome with the right evidence, education or design modification. However, any barrier that is discovered during a device's development and seen as absolute, such as a lack of a real clinical need, should raise question as to whether there is value in continuing with development or whether a change of focus or design may be required. POCT barriers may also evolve over time and a degree of 'future proofing' may be required. This is particularly true in respect to POCT for infectious diseases that may

**Box 8** Verbatim quotations to highlight the other (non-financial) resource use facilitators to POCT adoption (= Industry, CI = Clinician, Com = Commissioner)

"*a lot of appointments are taken up with people coming back for results, a lot of resources, text messages; if you've got the results there and then and they've got the results they don't need to phone back and people are satisfied, they go away with a printout"* (CI)
"*a bigger, more strategic view, and that is this whole issue of (…) trying to keep patients out of hospital*" (Com)
"*I am a patient in lots of things and I'd rather go and see my GP and then be able to say, "Well actually it's still a bit high, let's change it now," and that be done in the one visit than me going, me get a form, me have to go somewhere else to get my blood tests done*" (I)

be novel, mutate over time or develop resistance to antimicrobials. This is discussed further in a publication by European Joint Programming Initiative on Antimicrobial Resistance Transnational Working Group 'Antimicrobial Resistance-Rapid Diagnostic Tests' who propose their 'mix-and-match' package for the implementation of POCT.[16]

Each stakeholder group has different motivations for POCT adoption. Patient care is central to all; but clinicians may be influenced by their specialty, knowledge and the particular patients under their care, patients themselves respond to personal experiences, commissioners will likely have a defined focus on pathway benefits, as will regulators at a wider societal level and finally industry will be more focused on commercial aspects. In some circumstances patients have been shown to convey a more positive outlook regarding the adoption of POCT into healthcare systems compared with healthcare providers and industry.[17]

Such different motivations may explain the variation in barriers and facilitators identified by each group. While some were highlighted by most stakeholder groups (eg, cost of test, siloed budgets, patient experience or training requirements) many others were more frequently picked up by one particular group such as industry's frustration at the lack of clinician buy-in or commissioner's focus on reducing attendance at healthcare services, reconfiguration of pathways and facilitating local enhanced services. Reimbursement and funding mechanisms for new POCT will be test and setting specific and examples can be seen from the ways POCT testing for C reactive protein can be established in primary care[6] and of financial drivers influencing practice.[18] When looking at stakeholder motivations it may be more useful to describe the barriers and facilitators as incentives or disincentives to adoption as often the implementation of a POCT strategy will bring advantages to some stakeholders but disadvantage (usually financially) others. Furthermore, incentives and attitudes may be influenced further by culture and international differences have been demonstrated in the preferred attributes of POCT.[19]

This study supports and builds on the findings of the previous literature described in the introduction with the advantage of incorporating the diverse range of stakeholders included in the present study to allow for a comprehensive presentation of adoption influencers to be reported. The two most commonly cited barriers to POCT adoption in the current study were the cost of the test and associated training burden. This supports the results of the systematic review of Quinn *et al*[9] who reported the mostly commonly reported factors were related to economic issues, quality assurance and regulation. Quality assurance is a recurring concern in regards POCT where the end user may have limited or no training in the device itself or interpretation of the result.

When the emergent themes from this study are compared with the barriers identified in the interim report of the AAR[9] there is also good correlation. The themes

from the AAR were culture, data and evidence, finance and budgeting, systematic complexity and informational complaints, capacity/capability, dominance of the pharma paradigm and immaturity/uncertainty of landscape, all of which were highlighted by the stakeholders we recruited in the present study. Stakeholders frequently used the evaluation of pharmaceuticals as a comparator, usually envying their established methodology and large budgets for evidence generation and marketing. This was echoed in the AAR interim report, where pharmaceutical industry was described as dominating medical technology adoption given that National Institute for Health and Care Excellence (NICE) processes are based on pharmaceutical technology, horizon scanning is limited outside of pharmaceuticals and the challenging approval pathway for companion diagnostics.

This interview study highlighted several good examples of the use of POCTs but these tended to be small-scale pilot studies involving a small local practice or population. Furthermore, these were frequently short-term projects that were discontinued once funding came to an end, regardless of level of success. Such pilot schemes were rarely published or disseminated so other regions often have to reinvent POCT pathways leading to a repetitive evaluation processes and adding to the cost of establishing POCT services. This approach can be described as 'bottom-up' adoption and, with the one exception of faecal calprotectin, no examples were provided of 'top-down' adoption approach with central bodies driving adoption at a national level. Such 'top-down' adoption may be able to significantly increase the number of appropriate POCTs that are adopted following the rigorous generation of high-quality evidence of patient and service benefits. National bodies including the NIHR have recognised this approach and funded research groups to improve the evaluation of diagnostics to accelerate adoption. The NIHR Medtech in vitro diagnostics cooperatives[20] have multidisciplinary expertise and collaborate with industry and other stakeholders to analyse the impact of diagnostic to healthcare systems and generate data and evidence on a wide range of factors that contribute to the adoption of devices into clinical practice. The NICE has a health technology adoption team that aims to identify implementation barriers and promote technology adoption and includes diagnostic tests in its portfolio.[21] There are also examples of this approach being used by international consortiums.[22]

This study has limitations. Given the resources available for this study a minimum of eight participants was included in each stakeholder group. While the interviews were in depth and saturation appeared to have been reached increasing the number of participants may have improved the breadth of findings from the study. Recruitment strategy varied between stakeholder groups and aimed to provide a representative sample of participants but due to the time burden associated with conducting semistructured interviews a degree of convenience sampling was unavoidable and the study may be prone to recruitment bias. Furthermore, participants were often providing a perspective from their experiences within the UK NHS and it is unclear how transferable some of the barriers and facilitators identified would relate to different healthcare systems. One further limitation relating to this study is that the interviews were designed to develop the POCKET checklist with the identification of barriers and facilitators being undertaken as a secondary objective. This led to all POCTs being grouped together and many of the barriers and facilitators may be unique to a particular POCT testing strategy (such as self testing, primary care testing etc). However, relevant questions were included in the topic guide with a diverse group of stakeholders to comprehensively identify all the barriers and facilitators they have encountered with their expertise and experience.

This study provides an overview of the barriers and facilitators to the adoption of POCTs. In the most part, these can be seen as incentives and disincentives. Widespread adoption of POCT technology can only occur if the disincentives can be predicted and mitigated. Therefore, these factors should be addressed as early as possible in the device development pathway to maximise the likelihood of widespread adoption. Qualitative research methodology including clinical needs assessments and stakeholder analyses can be used to assist this process and should all be undertaken early are regularly reviewed throughout the device development pathway.

**Contributors** GBH, JH and JB conceived the project. GBH, JH, JB and MZN planned the project. JH undertook the data collection. JH and MZN analysed the data. GBH, JH, JB and MZN were involved in interpreting the data. All authors were involved in drafting the final manuscript.

**Funding** This research was supported by the National Institute for Health Research (NIHR) Diagnostic Evidence Co-operative London at Imperial College Healthcare NHS Trust.

**Disclaimer** The views expressed are those of the authors and not necessarily those of the NHS, the NIHR or the Department of Health.

**Competing interests** None declared.

**Patient consent for publication** Not required.

**Ethics approval** The protocol was approved by the Joint Research Compliance Office Imperial College, London, and the Imperial College Hospitals National Health Service (NHS) Trust Research and Development Department (ICREC References 14IC2186 and 14SM2190).

**Provenance and peer review** Not commissioned; externally peer reviewed.

**Data availability statement** No further data are available.

**ORCID iDs**
Jeremy R Huddy http://orcid.org/0000-0003-1966-3149
Melody Zhifang Ni http://orcid.org/0000-0001-7356-848X

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
