## [Reviewer comments · BMJ Open]

ARTICLE DETAILS

TITLE (PROVISIONAL)	A Qualitative Analysis of Stakeholder Interviews to Identify the Barriers and Facilitators to the Adoption of Point-of-Care Diagnostic Tests in the United Kingdom
AUTHORS	Huddy, Jeremy; Ni, Melody; Barlow, James; Hanna, George

VERSION 1 – REVIEW

REVIEWER	Donna Lecky Public Health England, Primary Care and Interventions Unit, England
REVIEW RETURNED	15-Aug-2020

GENERAL COMMENTS	Thank you for writing such a clear and concise manuscript on the barriers and facilitators to the adoption of Point-of-Care diagnostic tests. The originality of this work lies in the inclusion of views from a range of stakeholders, including industry, to POCTs in general. That being said, there have been numerous studies in this area and I would expect to see a broader comparison of findings to these studies in your discussion. I also have minor editing suggestions for improvement before it is published; see attached file and below: 1. Page 6 line 37 - remove the word meet2. When referencing other articles the authors names should not be in italics, please correct throughout3. Page 7 of 38, line 20 - undertaken has been used twice in one sentence, please rephrase4. Page 7 of 38, starting line 25 - Awkward phrasing in this sentence, perhaps change to Stakeholders included clinicians (laboratory staff...5. Page 7 of 38, line 35 - It's fairly standard practice to just have the initials in the text, you don't need to write 'by one author'. i.e. the interviews were carried out by JRH.6. Page 7 of 38, line 52 - Is this nurse an author on the paper or at least acknowledged by name for their contribution to the work?7. Results starting line 14 -These first 2 sentences could be simplified to Forty three stakeholders comprising regulators (10), Industry (9), commissioners (8), clinicians (8) and patients (8), were interviewed between August 2014 and December 2015.
--

	8. Results, line 33 - no need to have the origin of Skype named again 9. Page 9 of 38, line 18 - please add a comma after the word intervention 10. Page 21 of 38, line 16 - this sentence in its current format is quite difficult to read, please revise 11. Page 21 of 38, last line - please give a reference for the one example mentioned - The reviewer provided a marked copy with additional comments. Please contact the publisher for full details.
--	---

REVIEWER	John P. Hays Erasmus University Medical Centre Rotterdam (Erasmus MC)
REVIEW RETURNED	15-Sep-2020

GENERAL COMMENTS	The reviewer provided a marked copy with additional comments. Please contact the publisher for full details.
--

VERSION 1 – AUTHOR RESPONSE

Reviewer: 1

Reviewer comment: Thank you for writing such a clear and concise manuscript on the barriers and facilitators to the adoption of Point-of-Care diagnostic tests.

The originality of this work lies in the inclusion of views from a range of stakeholders, including industry, to POCTs in general. That being said, there have been numerous studies in this area and I would expect to see a broader comparison of findings to these studies in your discussion.

Response: Thank you for your feedback regarding our study. In keeping with both the reviewer's feedback a broader comparison to the existing literature has been incorporated throughout the manuscript.

Reviewer comment: I also have minor editing suggestions for improvement before it is published; see attached file and below:

1. Page 6 line 37 - remove the word meet
2. When referencing other articles the authors names should not be in italics, please correct throughout
3. Page 7 of 38, line 20 - undertaken has been used twice in one sentence, please rephrase
4. Page 7 of 38, starting line 25 - Awkward phrasing in this sentence, perhaps change to Stakeholders included clinicians (laboratory staff...

5. Page 7 of 38, line 35 - It's fairly standard practice to just have the initials in the text, you don't need to write 'by one author'. i.e. the interviews were carried out by JRH.

6. Page 7 of 38, line 52 - Is this nurse an author on the paper or at least acknowledged by name for their contribution to the work?

7. Results starting line 14 -These first 2 sentences could be simplified to

Forty three stakeholders comprising regulators (10), Industry (9), commissioners (8), clinicians (8) and patients (8), were interviewed between August 2014 and December 2015.

8. Results, line 33 - no need to have the origin of Skype named again

9. Page 9 of 38, line 18 - please add a comma after the word intervention

10. Page 21 of 38, line 16 - this sentence in its current format is quite difficult to read, please revise

11. Page 21 of 38, last line - please give a reference for the one example mentioned

Response: Thank you very much for the feedback regarding our manuscript. The editing suggestions you have provided have been amended in the manuscript.

In regards point 6, the research nurse who assisted with the analysis of the qualitative data has since left the department. She was invited to contribute to the preparation of this manuscript but declined and requested not to be named in the submission.

In regards point 11, this was a reference to the NICE's Health Technologies Adoption Programme for faecal calprotectin and a link to the NICE website for this as been added to the manuscript references.

Reviewer: 2

Reviewer comment: P2 / 3 Please change the title of the manuscript to "Barriers and Facilitators to the Adoption of Point-of-Care Diagnostic Tests in the National Health Service of the United Kingdom" or something similar.

Response: The title has been amended in keeping with your feedback and advice from the journal editorial team.

Reviewer comment: P4 / 20-25. This sentence suggests a total of 5 x 43 interviews, when in fact the total number of interviews for these 5 groups was only 43.

Response: This has been clarified in the abstract (page 3, line 9).

Reviewer comment: P4 / 31 - I think that text relating to: "1) the decision whether to treat or not, 2) more accurate decision making regarding the correct treatment to prescribe and 3) for home testing - whether to visit a healthcare professional (e.g. pharmacist or general practitioner) or not " should also be included as potential advantages of POCT testing.

Response: These advantages of POCT have been added to the introduction (page 4, lines 17-19).

Reviewer comment: P4 / 50 - 'for' is missing

Response: This has been amended in the manuscript.

Reviewer comment: P4 / 52 - Translational research may be somewhat different than the development of POCT diagnostics. POCT diagnostics may be based on new or already accepted

translational research. Further, the development of POCT diagnostic devices involves the development, manufacture and testing of the POCT device itself.

Response: We would agree with this statement. The evidence that is required to justify the adoption of POCT in healthcare systems in multi-faceted and was extensively studies in the development of the POCKET checklist (Clin Chem Lab Med. 2019 May 27;57(6):845-855). This has been referenced in the introduction and the text amended to include the importance of the point you have made (page 6, lines 3-6).

Reviewer comment: P6 / 24 Presumably the authors mean 'scaling up the adoption and development' of POCT technology rather than scaling up 'production capacity'. This should be made clear in the text.

Response: This has been clarified in the text (page 5, line 13).

Reviewer comment: P8 / 14 - The numbers interviewed are quite small and it may have been more appropriate to interview 43 stakeholders from a single group. Depth rather than breadth.

Response: The number of participants was based on the time and resources available for the study. The rationale for this was described in the initial protocol published in BMJ Open. A conscious decision was made to identify a wide variety of stakeholders for the study and gain a breadth of opinion. Saturation of data was apparent in all stakeholder groups.

Reviewer comment: P9 / 10 - The authors should define what they mean by "some POCTs".

Response: This sentence has been rewritten in the manuscript (page 8, lines 10-12).

Reviewer comment: P11 / 20 - Not just the successes, but also the failures.

Response: This has been clarified in the manuscript (page 10, lines 14-15).

Reviewer comment: P11 / 46-48 - Is this a statement of requirement or a statement of the problem from the commissioner?

Response: This is a statement of requirement. The commissioner was explaining that evidence regarding diagnostic accuracy is insufficient and there is a need to demonstrate that the test under evaluation has the ability to impact the clinical pathway (clinical utility).

Reviewer comment: P12 / 23 Please provide references for "Whilst this may be "perceived" or a "misperception"

Response: This description was taken from the interview transcripts. This has been clarified in the manuscript (page 11, line 12).

P13 / 3 - A good example of this is societal savings in terms of a reduction in the spread of infectious pathogens or antimicrobial resistance.

Response: Thank you, we have included this example as you suggest and provided a reference to illustrate this (page 12, lines 13-14).

Reviewer comment: P15 / 31-35. I would be interested in reading a comment on the new EU IVDR regulations and the BREXIT transition period (<https://www.gov.uk/guidance/medical-devices-eu-regulations-for-mdr-and-ivdr>).

Response: Thank you for highlighting this important example; the new regulations were frequently mentioned in the interviews. A reference for this has been included in the manuscript. The interview study was undertaken prior to the BREXIT referendum and a comment regarding this has been included in the text (page 14, lines 16-20).

Reviewer comment: P19 / 52 There are only a few references in the Discussion, which actually reads as a Conclusion and only describes the results of this study itself. Extra text in the Discussion should

be used to discuss the findings in the text boxes. All of these factors should, where/if possible, be linked to references. For example.

Response: Thank you for this feedback. We have added examples and text to the discussion to improve the comparison of our study with available evidence. Specific examples of this are given below.

Reviewer comment: Text Box 1. - Basically you are showing that industry is able to produce these tests, but that clinicians may be reticent to use them. A discussion point is not only differences in opinions, but also in knowledge and expectations of POCT, between medical microbiologist clinicians, non-microbiologist clinicians, general practitioners and patients.

Response: We agree this is an important discussion point. The second paragraph in the discussion focuses on this and has been amended.

Reviewer comment: Text Box 2. - Highlight, as an example, the discussion associated with NHS England/Boots and General Practitioners e.g. <https://bjgp.org/content/67/656/110.1>

Response: We agree this article adds a financial perspective and have included it in the manuscript discussion (page 20, lines 4-5).

Reviewer comment: Text Box 3 - Explain briefly the role of e.g. the NIHR Healthcare MedTech and In vitro Diagnostics Co-operative (<https://www.community.healthcare.mic.nihr.ac.uk/>), the National Institute for Health and Care Excellence (NICE <https://www.nice.org.uk/>) etc?

Response: The roles of these bodies have been included in the manuscript (page 21, lines 15-25).

Reviewer comment: Text Box 6 - I miss the efforts required to maintain the quality standards of POCT users (and registration of new users) over time. Also the development and/or adoption of existing external quality assurance schemes. For patients this is perhaps not so important, as they are unable prescribe drugs to themselves, but maintaining an independent quality chain (outside of the quality parameters determined and exercised by POCT developers) is important in maintaining POCT quality and POCT trust within hospital wards and general practitioner facilities.

Response: Thank you for highlighting this important point. It has been highlighted in the manuscript discussion (page 21, lines 18-19).

Reviewer comment: Text Box 8 - General practitioners being challenged in their decision making processes by a patient yielding a POCT result. Also, the possibility of patients potentially making appointments with their general practitioner, just because they don't believe a POCT result, or cannot clearly interpret the read-out of a used POCT test.

Response: We agree these are potential scenarios that may arise from the introduction of POCT. However, these examples were not put forward by the stakeholders included in the interview study and therefore have not been included in the manuscript.

Reviewer comment: N.B. Also, from this reviewer's own research collaborations (there are many other publications available on these issues and these publications are provided as potential examples), potentially interesting notes for the discussion concepts include, not exclusively, as the "Mix and Match" approach (<https://doi.org/10.1007/s10096-019-03492-4>); the disparity between clinician and patient view of POCT doi: 10.2147/PPA.S44889.) and specific to AMR POCT, but with issues relevant to all POCT developers (<https://doi.org/10.1038/s41579-018-0098-9>).

Response: Thank you for suggesting these relevant examples.

Reviewer comment: P21 / 33-34 - In fact, I would suggest that pharmaceutical companies are very choosy in the drugs that they are prepared to invest in. With the lowest risk to reward ratio likely being drugs that are required to be taken regularly and for chronic conditions. It

may be that successful POCT development also needs to abide by these rules in order to be successful.

Response: Thank you for this excellent point. Whilst we would agree with it, our study has not included any data to support this statement so we have not included this point in the manuscript.

Reviewer comment: P21 / 59 - 60 - The adoption of 'universally' accepted Target Product Profiles e.g.; <https://www.who.int/research-observatory/analyses/tpp/en/> and Procurement Platforms e.g. <https://www.karolinska.se/en/karolinska-university-hospital/Innovation/pippi/> may be useful here.

Response: Thank you for sharing the link to this consortium. We have included a reference to this group in the discussion.

Reviewer comment: P22 / 14 - Perhaps the most obvious limitation is the low numbers of respondents?

Response: The following sentence has been included in the discussion of limitations. "Given the resources available for this study a minimum of eight participants was included in each stakeholder group. Whilst the interviews were in depth and saturation appeared to have been reached increasing the number of participants may have improved the breadth of findings from the study." (page 21 line 26 – page 22, line 1).

Reviewer comment: P22 / 45 - Due to the low number of respondents and its focus on the UK, I would be reticent to accept that this study provides a 'comprehensive overview'.

Response: We have removed the word "comprehensive" from this sentence giving the focus of the study on the United Kingdom.

Reviewer comment: P22 / 56 - The authors are referred to, as an example, <https://doi.org/10.1007/s40258-019-00516-0>, which used a Discrete Choice Experiment (DC) survey and interviewed 988 respondents. Although relating to AMR, the technique used and conclusions are likely to be applicable for patient preferences for all POC diagnostics.

Response: Thank you for highlighting this article. It has been added to the discussion.

Reviewer comment: P23 / 56 - Reference 8. There is no URL for this (online) reference.

Response: Thank you for highlighting this omission. A link to the interim report has now been included.

Reviewer comment: P35 - Appendix. Was the same POCKET interview used for all stakeholder groups? Did the authors check that the respondents understood all of the terminology used in the questionnaire? For example, competition analysis, decision analysis, diagnostic accuracy study?

Response: The semi-structured interview methodology allowed some flexibility and so interview questions were tailored to the role and level of experience of the interviewee. Terminology was explained where required. This part of the methodology was outlined in the original protocol manuscript and has now been highlighted in the current paper.

VERSION 2 – REVIEW

REVIEWER	John P. Hays Erasmus University Medical Centre (Erasmus MC), the Netherlands
REVIEW RETURNED	30-Jan-2021
GENERAL COMMENTS	Good luck with the submission.